# Relevance of the Adjuvant Effect between Cellular Homeostasis and Resistance to Antibiotics in Gram-Negative Bacteria with Pathogenic Capacity: A Study of *Klebsiella pneumoniae*

**DOI:** 10.3390/antibiotics13060490

**Published:** 2024-05-26

**Authors:** Mildred Azucena Rivera-Galindo, Félix Aguirre-Garrido, Ulises Garza-Ramos, José Geiser Villavicencio-Pulido, Francisco José Fernández Perrino, Marcos López-Pérez

**Affiliations:** 1Doctorado en Ciencias Biológicas y de la Salud Universidad Autónoma Metropolitana, Ciudad de México, México Universidad Autónoma Metropolitana-Unidad Xochimilco Calz, del Hueso 1100, Coapa, Villa Quietud, Coyoacán CP 04960, Mexico; mildred.tnt@gmail.com; 2Environmental Sciences Department, Division of Biological and Health Sciences, Autonomous Metropolitan University (Lerma Unit), Av. de las Garzas N◦ 10, Col. El Panteón, Lerma de Villada CP 52005, Mexico; j.aguirre@correo.ler.uam.mx (F.A.-G.); j.villavicencio@correo.ler.uam.mx (J.G.V.-P.); 3Centro de Investigación Sobre Enfermedades Infecciosas (CISEI), Instituto Nacional de Salud Pública (INSP), Cuernavaca CP 62100, Mexico; ulises.garza@insp.mx; 4Department of Biotechnology, Division of Biological and Health Sciences, Universidad Autónoma Metropolitana-Unidad Iztapalapa, Av. San Rafael Atlixco 186, Leyes de Reforma, México City CP 09340, Mexico; fjfp@xanum.uam.mx

**Keywords:** adjuvant effect, antibiotic resistance, drug development, homeostatic regulation, *Klebsiella pneumoniae*

## Abstract

Antibiotic resistance has become a global issue. The most significant risk is the acquisition of these mechanisms by pathogenic bacteria, which can have a severe clinical impact and pose a public health risk. This problem assumes that bacterial fitness is a constant phenomenon and should be approached from an evolutionary perspective to develop the most appropriate and effective strategies to contain the emergence of strains with pathogenic potential. Resistance mechanisms can be understood as adaptive processes to stressful conditions. This review examines the relevance of homeostatic regulatory mechanisms in antimicrobial resistance mechanisms. We focus on the interactions in the cellular physiology of pathogenic bacteria, particularly Gram-negative bacteria, and specifically *Klebsiella pneumoniae*. From a clinical research perspective, understanding these interactions is crucial for comprehensively understanding the phenomenon of resistance and developing more effective drugs and treatments to limit or attenuate bacterial sepsis, since the most conserved adjuvant phenomena in bacterial physiology has turned out to be more optimized and, therefore, more susceptible to alterations due to pharmacological action.

## 1. Introduction

Resistance mechanisms must be understood as continuous adaptation processes in the natural history of bacteria [1,2]. Resistance mechanisms and other cellular functions related to antibiotics are intricate in environmental responses to changes in ecosystem conditions. These ancient mechanisms are highly optimized in bacterial physiology [3]. Bacterial resistance is a significant concern in healthcare settings globally and is continuously evolving. Therefore, it is crucial to comprehend the mechanisms that underlie this problem and ask relevant research questions that aim to develop effective strategies and treatments to combat it. One mechanism that has been demonstrated to contribute to antibiotic resistance in bacteria is the alteration of cellular homeostasis [4]. Cellular homeostasis is the ability of cells to maintain a stable internal environment despite fluctuations in external conditions. This balance is crucial for the proper functioning of cells and the overall survival of microorganisms. 

In the context of antibiotic resistance, cellular homeostasis plays a crucial role in resisting the action of antimicrobials. Homeostasis ensures that resistant cells continue to function effectively in the presence of antimicrobials. This guarantees the proper functioning of essential metabolic processes, such as DNA replication, protein synthesis, and energy production [4,5]. Understanding the molecular basis of *Klebsiella pneumoniae* is important due to its association with healthcare-associated infections (HAIs), its adaptive mechanisms, and its impact on antibiotic resistance [6]. Several studies have shown that alterations in metabolic pathways can improve survival and antibiotic resistance in cellular homeostasis [7]. These modifications may also result in the activation or overexpression of efflux pumps, which are significant resistance mechanisms in bacteria [8]. The acquisition of antibiotic resistance mechanisms is typically viewed as a response to environmental factors rather than a fundamental aspect of bacterial metabolism [9]. Recent research has shown that intrinsic resistance in pathogenic bacteria requires the concerted action of many genes and genetic circuits. Several of these genes and circuits play relevant roles in *K. pneumoniae* [10]. The most conspicuous studies in this sense refer to strains that have resistance to multiple antibiotics in clinical settings; these strains usually have alterations in their metabolic networks, including components of the electron transport chain and the metabolism of amino acids, fatty acids and nucleotides [11]. These modifications not only contribute to resistance but also form the basis for explaining the adaptive processes involved in colonizing environments with intense selection pressure due to the presence of antimicrobials. 

This enabled us to infer a close relationship between homeostatic regulation mechanisms and the global impact on the gene regulatory network (GRN) of *K. pneumoniae*. 

## 2. Bacterial Homeostasis Mechanisms

Cellular homeostasis is a fundamental characteristic of all organisms. It enables them to focus on their physiology, adapt to a variety of environments, and ultimately survive and reproduce. This is one of the basic physiological principles. Microorganisms have a high area/volume ratio, making them vulnerable to rapid changes in the physicochemical and biological parameters of the ecosystem. Their evolutionary history has optimized their environmental perception systems to respond quickly to these changes; these conclusions are supported by various studies [12,13,14]. These mechanisms have a comprehensive and significant impact on the GRN, modifying various aspects of metabolism, transport, and response to environmental stimuli. One of the homeostatic regulation mechanisms relevant to the focus of this study is limited to the regulation of energy production [15], particularly changes in the electron transport chain that occurs as a consequence of increased energy demand to cope with a stressful ecosystem context (Figure 1). Under this scenario, a decrease in membrane potential may occur. This has been reported to affect the absorption process of certain antibiotics whose entry into the cell interior is conditioned by the proton–motive force [16]. Furthermore, certain metabolic byproducts produced by alterations in biochemical pathways, specifically those associated with amino acids, have been found to interfere with the activity of antibiotics. This has been reported in several relevant studies [17,18,19,20]. In contrast, transcription factors that act as global metabolic regulators have been linked to bacterial physiological states that make them more susceptible to antimicrobials. An example of this type of homeostatic mechanism is the Crc factor (Global Carbon Metabolism Regulator) [20,21]. Processes associated with the regulation of this transcription factor have been linked to dormancy or biofilm formation, which can affect bacterial susceptibility to antibiotics. In a dormant state, bacteria are in a physiological state that is less susceptible to the action of antibiotics because their transport function and main mechanisms of molecular genetics are limited, such as replication and its consequent effect on growth, which is arrested. Under these conditions, bacterial populations can survive the presence of high antibiotic concentrations [22]. Specifically, several studies have investigated the process of reversing the dormancy state in non-spore-forming bacteria using *K. pneumoniae* [23]. Another mechanism for homeostatic regulation involves bacterial association, specifically the formation of biofilms and their relevance to antibiotic resistance. This topic has been extensively reported in the scientific literature, particularly for *K. pneumoniae*; several studies have emphasized the importance of this cellular aggregate configuration in acquiring the resistance phenotype [24].

### 2.1. Nonspecific Mechanism and Biofilm Formation

Figure 2 summarizes the impact of biofilms on non-specific adaptation to antimicrobials. Studies have shown that cell aggregation and biofilm formation aid in the adaptation to bacterial communities under various stress conditions [25]. 

This refers to the impact that biofilm growth has on fundamental processes. It has been demonstrated that bacterial physiology in biofilm conditions results in a decreased growth rate and limited uptake of nutrients compared to bacterial populations that grow in planktonic form [26]. Other studies have referred to the ‘impermeability’ of these structures, which is directly related to their effect on transfer and transport phenomena [27]. Figure 2 demonstrates how bacterial capsules and polymeric substances in biofilms facilitate the retention of antibiotics. It is important to briefly explain the adequacy of cell aggregation in a non-specific manner. When cells form groups or large groups, a glycocalyx is also formed that surrounds the aggregate not only peripherally, but also forms an internal matrix. Under these conditions, there are cells arranged peripherally and others in the interior, so an enormous variety of compounds, from secondary metabolites of diverse nature to antibiotics, which present more frequently in bacteria that inhabit the soil, are retained by the peripheral barrier constituted by glycocalyx and bacterial cells, generating a concentration gradient that decreases towards the interior or nucleus of the aggregate. These low concentrations can reach sublethal levels, facilitating favorable conditions for cells to adapt. Maintaining these selection pressure conditions can lead to resistance, as cells direct their energy flow to other areas and functions, and reproduce [28].

Several studies have described these conditions in detail regarding *K. pneumoniae* [29,30] and other microorganisms [31,32]. In contrast, research has shown that bacterial biofilms, which are complex microbial communities enclosed in a matrix of exopolysaccharides, contain subpopulations with varying metabolic activities, including inactive states that enhance their tolerance to antibiotics [33]. This topic has been extensively researched in *K. pneumoniae* [24,34]. Another important pathway associated with cellular homeostasis, although not exclusive to *K. pneumoniae*, is the regulation of the carbon cycle, which is essential for energy production. It is crucial to regulate the expression of genes involved in aerobic and anaerobic respiration, fermentation, and the biosynthesis of amino acids and nucleic acids to maintain homeostatic balance. This also includes responding to heat shock, osmotic shock, and oxidative stress [4,8,20]. Furthermore, homeostatic regulation involves maintaining an acid–base and osmotic balance through ion channels, pumps, and transporters. These mechanisms help keep intracellular pH within an optimal range and regulate ion movement across the cell membrane. Bacteria can accumulate compatible osmolytes, such as trehalose, to counteract osmotic shock and maintain cellular homeostasis [35]. In this sense, the genes *otsA* and *otsB* encode the enzymes trehalose-6-phosphate synthase (otsA) and trehalose-6-phosphate phosphatase (otsB), respectively, which are associated with the synthesis of trehalose from glucose-6-phosphate, and their expression is regulated in response to osmotic stress signals [36]. 

### 2.2. Stressful Ecosystem Conditions and Adaptive Response

Additionally, it is important to mention the utilization of various stress response pathways that aid in coping with changes in the external environment. These pathways can be triggered by a range of stressors, including alterations in temperature, pH, and osmolarity, and can help adapt to these changes by altering their cellular physiology and gene expression patterns [20]. A well-described example in the literature is associated with a two-component system that has been published in several studies [35,37]. The system comprises two primary elements. The first is (1) a signal-sensing protein located in the cell membrane that detects specific environmental stimuli, such as the presence of nutrients, oxygen concentration, changes in pH, and temperature, and the presence of toxic chemicals [35]. When activated, this system transfers a phosphate group from an ATP molecule to the response regulator protein. (2) The second component refers to a response protein, known as a response regulator, located in the cytoplasm of the cell. This protein responds to the phosphate transferred by the signal sensor protein and triggers a series of adaptive responses in the cell, including the regulation of gene expression, the alteration of metabolism, and the modification of the cell surface. *Klebsiella* has several two-component systems, such as the PhoP-PhoQ system that responds to phosphate deficiency, and the KvgAS system that is associated with antimicrobial resistance [38]. Other studies have investigated *K. pneumoniae* in adaptive analyses, particularly under resistant conditions. Ref. [39] includes a study that is very significant because it offers a unique perspective for new research horizons. It explores the connections between genetic circuits and selection pressure from antibiotics, which can lead to the development of new approaches that block multiple pathways, making it harder for pathogenic bacteria to adapt. 

In general, this system can adapt to changes in temperature or the pH of the surrounding environment by altering its gene expression or metabolic activity. Similarly, when nutrients are scarce, it can activate genes involved in nutrient acquisition while repressing genes involved in cell growth and division. Additionally, exposure to toxins or other harmful substances can activate mechanisms to detoxify or eliminate harmful compounds [40,41]. Genetically, nonspecific adequacy mechanisms have a more widespread impact on GRN, since growth physiology must be reoriented, first affecting metabolic energy flows [42]. Other studies have focused on regulating intracellular molar concentrations, which affect the mechanisms of gene expression through transduction chains [43]. Some studies have focused on this aspect of homeostasis associated with resistance in *Pseudomonas* [44]. The fitness phenomena were related to homeostasis and resistance to antibiotics transition from an initial phase of previous bacteriostasis towards a resistance phenotype that varies depending on the time interval in which the selection pressure due to the presence of the antibiotic is maintained. This can be inferred from different published studies [45,46]. The adaptation process can be approached with a pyramidal vision by relating genetic circuits that have more generalized actions in cellular physiology in the GRN, such as those associated with homeostatic regulation, to more peripheral genetic circuits in the GRN that have much more specific actions. The activation of more specific resistance mechanisms can be associated with the gene regulation phenomena related to cell aggregation and substrate anchoring, such as quorum sensing (QS). This thesis has already been studied [47]. These scenarios are particularly relevant because they explore the complexity of resistance mechanisms connected to the number of genes involved, and a direct relationship cannot be established between the presence of a specific gene and resistance to an antimicrobial, in particular. 

This statement suggests that the emergence of the resistance phenotype is due to an adjuvant phenomenon between more general circuits, which do not have a direct effect on the antimicrobial, and more peripheral circuits in the GRN, which directly affect the antimicrobial. This reasoning implies that it is an adjuvant phenomenon between more general circuits (that do not have a direct action on the antimicrobial) and more peripheral circuits in the GRN (that directly affect the antimicrobial), which contribute to the emergence of the resistance phenotype.

## 3. *Klebsiella pneumoniae*: Adequacy and Resistance Phenotype

The persistence of microorganisms in their natural environment is based on their adaptability and plasticity in different ecological niches. The adaptation and survival capacity of *Klebsiella* strains is one of the main reasons why this genus is widely distributed and, on the other hand, has managed to incorporate enormous diversity. *K. pneumoniae* is a Gram-negative bacillus that belongs to the Enterobacterales family. It is considered one of the species that has developed the most resistance over time [48]. The field of public health has shown great interest in this topic. *K. pneumoniae* resides in the human gastrointestinal tract as part of the normal microbiota and has a non-pathogenic phenotype. However, under certain circumstances, its physiology enables it to transition to a pathogenic phenotype. It is important to note that this transition is not a guaranteed outcome [49]. This physiological adaptability has been extensively studied [50,51] due to its relevance in the hospital context, where it has been reported as one of the main pathogens causing various types of nosocomial sepsis, particularly in hospitalized or immunocompromised patients [52]. 

A significant aspect of this physiological adaptability is closely linked to the relatively extensive genome size (~5–6 Mbp). In *K. pneumoniae* genomes, analysis identified that gene accumulation revealed an open pangenome [53]. This allows *K. pneumoniae* to acquire and maintain genes that code for specific functions, such as antibiotic resistance, virulence, and adaptation. *K. pneumoniae* can also acquire resistance genes from various sources, including plasmids and mobile genetic elements [9]. *K. pneumoniae* can utilize a diverse range of substrates as both a carbon and energy source. Additionally, it can fix atmospheric nitrogen, which is crucial for its survival in environments where nitrogen is scarce [54]. It can also withstand unfavorable conditions, such as fluctuations in pH and temperature [55]. Studies have investigated the mechanisms of adaptation to salinity [56] and other environments, such as the gastrointestinal tract [57]. Furthermore, it would be worthwhile to investigate the adhesion and colonization capabilities of *K. pneumoniae* on different biotic and abiotic surfaces, including catheters and medical equipment [58]. This has significant implications for the risk of hospital-acquired infections. This mechanism has been studied in the expression of genes related to capsule adhesion and formation. These genes facilitate the colonization of different tissues and surfaces [59]. In this sense it has been described that metabolic adaptations in *K. pneumoniae* may provide a selective advantage in the presence of antibiotics, allowing bacteria to survive and proliferate [60]. It has been reported that *K. pneumoniae* can adjust its metabolic pathways to counteract the effects of antibiotics by overexpressing mechanisms such as efflux pumps. These pumps actively remove antibiotics from the cell before they can affect the physiology of bacteria [61,62]. 

The importance of *K. pneumoniae* in the public health field is supported by several critical factors [63] such as hospital-acquired infections, high rates of antimicrobial resistance, spread, virulence, and genomic diversity, which have a significant economic impact due to prolonged hospital stays, treatment, and increased morbidity and mortality [64]. When considering the relationship between homeostatic regulation mechanisms and antibiotic resistance, it is important to focus on the regulation of the gene expression associated with the perception and response to changes in ecosystem conditions. Studies have focused on transcription factors, both repressors and activators, that are associated with genes such as *KbvR*. These factors regulate the production of external cell envelopes, such as the capsule or the external membrane, as well as central processes in cellular physiology, such as protein synthesis. Another factor, RamA, regulates different transport-associated genes that are exerted by efflux pumps [62]. To understand the adequacy of phenomena under stressful conditions, such as the uptake of nutrients or trace elements, it is fundamental to comprehend the conditioning of general aspects of *Klebsiella* GRN by these genes [65,66]. Studies have explored the metabolic network of *K. pneumoniae* in the context of resistance to multiple antimicrobials, with a specific focus on metabolic and transcriptional regulation [67,68]. These studies highlight the importance of identifying nodes in the *K. pneumoniae* genomic network, which may lead to new research opportunities to further investigate the processes of adaptation and resistance emergence in *K. pneumoniae*. 

## 4. Molecular Basis and Physiological Principles of Antimicrobial Resistance Mechanisms in *K. pneumoniae*

Antibiotic resistance in *K. pneumoniae* is worsened by selection factors such as the excessive and inappropriate use of antibiotics in human and veterinary medicine, as well as the spread of resistant strains in various environments [69,70]. This issue has significant implications, not only at the clinical level but also at the environmental and economic levels [26], compromising the ability to treat common and life-threatening bacterial sepsis. Additionally, infections caused by carbapenem-resistant *K. pneumoniae* strains are associated with high mortality rates and limited treatment options, and also facilitate their spread [64]. The scientific literature has delved into *K. pneumoniae* strains that carry resistance mechanisms to carbapenem antimicrobials. These strains are especially relevant since they are the main etiological agent of many nosocomial diseases. In this sense, it is pertinent to refer to several works that have specifically delved into the pathogenic physiology and resistance mechanisms of these strains [30,39]. On the other hand, it is pertinent to emphasize that these resistance mechanisms, based on enzymes as the carbapenemases, which are functionally β lactamases, affect the bacterial cell wall synthesis process and, with its bacteriostatic effect, prevent the growth of bacteria in saprophytic and pathogenic conditions. These mechanisms act under the regulation of more interconnected nodes in the GRN, and carbapenemases act more peripherally. In this sense, it is pertinent to delve deeper into the mechanisms that regulate these resistance mechanisms. 

Antimicrobial resistance caused by enzymes in *K. pneumoniae* is a significant public health concern. It should be viewed as an adaptive response of the microorganism to counteract a stressful ecosystem context, specifically limited to the effect of antibiotics. This resistance can be understood as a coupled perception–induction system. Enzyme-based resistance involves a process in which bacteria have an environmental perception system based on a specific reception mechanism for the detection of the presence of the stress agent (antibiotic). The perception of afferent information triggers the production of specific enzymes or other resistance mechanisms. This response system is interconnected and coordinated, resulting in not only a specific enzymatic action on the antimicrobial but also the regulation of the expression of these systems, which is conditioned by the concentration of the antibiotic [71,72]. Under these *K. pneumoniae* conditions, it generates very efficient resistance to the stressor [9,10,44,60]. It is important to note that the locations of these resistance genes have been detected in both the bacterial genome and mobile genetic elements, such as plasmids [9]. 

Regarding the efficiency of the response, it is important to note that this mechanism is highly regulated. Bacteria have control mechanisms that activate these genes only when necessary, as previously reported for *K. pneumoniae* [73]. This ensures that the production of enzymes and other resistance mechanisms is adjusted to the specific environmental conditions. Once activated, these genes prompt bacteria to produce enzymes that efficiently counteract the antibiotic’s effects by degrading or inactivating it. This allows the bacteria to direct energy towards reproduction even in the presence of the antibiotic. In *K. pneumoniae* and other microorganisms, there are regulons that control the expression of antibiotic resistance genes such as the *MarA* regulon, which is a regulatory protein that can activate the expression of efflux pump genes (such as AcdAB-TolC) and enzymes for antibiotic degradation, which can, in turn, be activated in response to the stress induced by antibiotics [74]. The *SoxS* regulon is an example that plays a crucial role in antimicrobial resistance, oxidative stress, and other environmental stressors [75]. Together with *MarA*, it contributes to the response to multiple stressors, including antibiotics. It regulates genes that encode efflux pumps, membrane proteins, and other factors that enable the microorganism to expel the antibiotic, resulting in a resistance phenotype [76]. The activation of these regulons and the overexpression of the resistance genes they control have significant clinical implications in the treatment of *K. pneumoniae* infections [77]. 

The *K. pneumoniae* literature on resistance mechanisms includes a group of mechanisms related to the regulation of RNA expression that are of special relevance. Dersch et al. conducted important studies on a regulatory system associated with sRNAs. These regulators operate at the post-transcriptional level by conditioning the expression of resistance genes through interaction with mRNA regions [78]. Other studies have investigated the regulation of gene expression in response to environmental factors, specifically focusing on RNAs involved in the activation of the QS system [79]. Additionally, a study has linked metabolic flux, specifically the production of 2,3-butanediol and acetoin, to the induction of the QS state [80]. Another study was conducted to investigate the proteomes of the polymyxin-resistant and polymyxin-sensitive strains of *K. pneumoniae* associated with *CrrAB, PmrAB, PhoPQ,* and *ArnBCADT* [81]. The study found significant differences in bacterial metabolism, emphasizing the importance of metabolic regulation in the mediation of antibiotic resistance. However, studies indicate that environmental sources can serve as a reservoir of resistance genes in various species. 

Therefore, it is crucial to monitor the prevalence of antibiotic resistance in *Klebsiella* and other bacterial species and take measures to prevent their spread whenever possible [82]. Finally, it is important to mention that antibiotics can also act as secondary cellular messengers. Their presence in the environment can trigger responses from bacterial cells that go beyond the simple action of killing or inhibiting growth [83]. 

## 5. The Impact of Cellular Homeostasis on *K. pneumoniae* Bacterial Resistance

Currently, there is a wealth of scientific literature focusing on the various aspects of homeostatic regulation and resistance. The following sections discuss some of the most relevant aspects in which the contributing effects can be measured. 

### 5.1. Cellular Efflux 

Cellular efflux significantly affects homeostatic regulation. Bacterial cells actively expel substances with varying toxic potentials from the cytoplasm to the outside, including antibiotics, secondary metabolites, heavy metals, and other toxic compounds. This causes a decrease in their concentration or elimination, thereby attenuating their physiological effect. Cellular efflux is primarily based on the transport of proteins embedded in the membrane. It can be divided into two categories based on specificity. (a) The more specific the efflux mechanism, the more efficient it is in interacting with the ligand and expelling it. However, these mechanisms require the recognition and regulation of expression, which means that the cell must have the necessary genes and regulatory mechanisms in its genome [84]. Non-specific systems have also been described [85]. (b) These mechanisms have been extensively studied in *K. pneumoniae* and have greater promiscuity with a wider range of ligands, allowing environmental adaptation under different conditions. However, the pathways for regulating the expression of these systems are more general and do not incur additional expenses. In this second category of cellular efflux proteins, they are described as non-specific factors that have contributed to the emergence of antimicrobial resistance in *K. pneumoniae* and other bacteria. Transportation through porins is also believed to be influenced by cellular efflux. An example of this are aquaporins, which are proteins found in the cell membrane that act as water channels, allowing for selective and rapid passage through the membrane. Aquaporins are present in a wide range of organisms, from bacteria to humans, and are essential for various biological processes. These processes include the regulation of cell volume, which in turn regulates molar concentrations, as well as the excretion of waste and absorption of nutrients [86]. This function is important because the concentrations of transcription factors, substrates, and secondary metabolites have been linked to various cellular response mechanisms in bacteria [87] and other organisms [88]. In *Klebsiella*, antibiotic efflux systems are regulated by a series of genes that control the production and activity of transport proteins. The regulation of these genes is complex and coordinated, involving transcriptional regulation proteins such as MarA, SoxS, Rob, and the response regulator NarL. NarL is sensitive to the presence of oxygen and nitrates in the environment [11]. 

Recently, studies have been published linking transporters associated with cellular efflux to multi-resistance. For example, Yi et al. worked with the AcrB transporter, and molecular dynamics studies demonstrated its significant effect on resistance to tetracycline and the survival of cefoxitin [89]. Other recent studies have investigated the role of the AcrAB operon in the phenomena of virulence and antimicrobial resistance in *K. pneumoniae* using strains with isogenic knockouts [61]. The OqxAB transporter has also been analyzed in other studies [90,91]. Furthermore, several studies on *K. pneumoniae* have made significant efforts to elucidate the role of cellular efflux in resistance, including the transporters of the RND group [92]. Finally, some relevant studies have referred to transporters of the Major Facilitator Superfamily (MFS) group. Specifically, the KpnGH transporter has been shown to play a significant role in resistance mechanisms to various antibiotics, including azithromycin, ceftazidime, ciprofloxacin, ertapenem, erythromycin, gentamicin, imipenem, ticarcillin, norfloxacin, polymyxin B, piperacillin, spectinomycin, tobramycin, and streptomycin [93]. It is important to note that several reviews have specifically focused on *K. pneumoniae* and its efflux systems associated with antimicrobial resistance [94]. 

### 5.2. Heat Shock Proteins

*Klebsiella* can grow at a wide range of temperatures. However, this ability may be dependent on its adaptation to environmental conditions [95,96]. Bacteria, including *Klebsiella*, can produce heat shock proteins, such as chaperones of the heat shock protein (HSP) family, which enable them to survive extreme temperatures or temperatures outside of the range in which *K. pneumoniae* typically thrives [97]. Heat shock proteins (HSPs) are a group of highly conserved proteins that are expressed in response to stressful situations, such as heat. They play a crucial role in protecting proteins from potential damage, which directly affects homeostatic regulatory functions [97]. When interpreting the presence of HSP in these bacteria with a homeostatic approach, it is important to consider that many bacteria with pathogenic potential may transition from free-living saprophytic forms in the environment to ecosystems associated with organisms with homeostatic regulation. In these ecosystems, stable conditions are maintained, and the bacteria are exposed to less temperature variation. In both cases, these proteins confer stability to the proteins they frequently interact with as chaperones, as previously published [98]. In the scientific literature, it is common to establish correlations between the mechanisms of adaptation to the stress caused by temperature and the mechanisms of resistance to different drugs. Some studies have even extended beyond microorganisms to focus on multicellular organisms. Cruz-Loya et al. demonstrated that in *E. coli*, heat shock and cold shock genes are induced by antibiotics. This suggests that some adaptation mechanisms at temperature are coupled to antibiotic resistance [99]. A study investigated the correlation between *Streptococcus pneumoniae* and resistance to β-lactam antibiotics. The study found that the heat shock protein ClpL is related to this resistance [100]. More recent studies have analyzed the effect of different antibiotics and exposure to different temperature ranges on microbial physiology, specifically growth. These studies are significant because they explore the evolutionary and adaptive implications of antibiotics on bacterial populations in a new way [101]. Furthermore, to demonstrate the common origin of these mechanisms, it is relevant to reference studies in eukaryotes. These studies establish correlations between temperature adaptation mechanisms and the evasion or attenuation of various drugs, for instance, studies related to breast cancer [102] or the pathophysiology of human diseases [103]. In *K. pneumoniae*, there is a scarcity of literature that relates these two mechanisms. However, some studies have explored the presence of chaperones and their relation to virulence [104]. Furthermore, studies have shown that the susceptibility of *K. pneumoniae* to aminoglycosides can be affected by heat shock treatments [105]. 

### 5.3. Response to O_2_ Availability and Reactive Oxygen Species 

*Klebsiella’s* pathogenic phenotype is mainly found in lung tissue, making it strongly aerobic and highly dependent on oxygen saturation. Oxygen serves as the final electron acceptor in the respiratory chain, and any restriction in this transfer leads to a decrease in the amount of ATP that bacteria can produce [106]. The regulation of the response to oxygen availability in *Klebsiella* occurs at the gene expression level through specific transcription factors. One of the primary regulators is the transcription factor FNR (fumarate nitrate regulatory factor), which functions as an oxygen sensor [107]. FNR binds to DNA in the presence of limited oxygen and activates the expression of the genes necessary for adaptation to conditions of low oxygen availability. Additionally, the global regulator CRP, which is also a key regulator in the response to glucose availability, plays a role in regulating the response to oxygen availability [108]. This gene has also been associated with the QS system. Therefore, we can consider its potential relevance in terms of resistance phenomena [109]. In contrast, research has shown that high oxygen concentrations can promote the production of ROS [110]. Therefore, it can be concluded that there is a correlation between oxygen exposure and intracellular ROS concentration. It is important to include studies that have examined the correlation between ROS and processes such as virulence and resistance in *K. pneumoniae*. This is because there is a thesis that suggests that the internal concentration of ROS affects adaptive physiology and bacterial homeostatic regulation [111,112]. 

A study carried out with *K. pneumoniae* found that the homeostatic regulation mechanism associated with the presence of ROS involves three transcriptional regulators: SoxS, SoxR, and OxyR. The study also determined a correlation with cellular efflux mechanisms [75]. Another relevant study investigated the correlation between bacterial oxidative stress and mutagenesis, as well as the repair processes in DNA. The study directly addressed the issue of bacterial adaptation to various environmental stressors, including the presence of antibiotics. The results showed a strong correlation between oxidative stress and adaptation through induced mutation [113]. Finally, another study analyzed the bactericidal effect of eugenol on carbapenem-resistant strains of *K. pneumoniae* by inducing ROS in *K. pneumoniae* [114]. This is an example of research aimed at attenuating or limiting resistance by affecting the cellular signaling pathway associated with homeostasis. 

### 5.4. Quorum Sensing 

Quorum sensing (QS) is the physiological state that bacteria reach after induction. This state is perceived by different environmental elicitors, which generally modify the bacterial phenotype, facilitating agglutination, flocculation, or the grouping of bacteria in liquid media and on surfaces. This phenotype is achieved by modifying and adapting the mechanisms that regulate genetic expression, starting with the perception of microorganism density that may be associated with the environment (Figure 3) [115]. This mechanism modifies the adaptive response of bacteria and is, therefore, one of the most relevant homeostatic regulation mechanisms. Microorganisms release chemical signaling molecules called autoinducers, which increase their concentrations based on their density. Cellular organization related to QS involves producing and detecting autoinducers, expressing QS receptors in the cell membrane, and activating genes associated with communication and adaptation to the environment [116]. In *Klebsiella*, QS is associated with regulating virulence and producing extracellular factors, including siderophores, digestive enzymes, and biofilms [117]. Antibiotics can impact gene expression if they are regulated by quorum sensing and if the antibiotic affects the production or detection of the autoinducers necessary for their regulation [118]. Some antibiotics have been shown to downregulate the production of autoinducers, affecting the regulation of QS and the expression of related genes [119]. Other antibiotics, such as aminoglycosides, have been shown to increase autoinducer production. As a result, they increase the expression of the genes regulated by QS, such as in *Vibrio cholerae* [120]. This is an interesting case because resistance is mediated by the induction of chaperones associated with heat shock. Therefore, it is an example of the confluence of two homeostatic regulation mechanisms contributing to the adaptive process. Specific genes that encode the enzymes involved in the synthesis of glycocalyx components have been identified in *Klebsiella*. The *wzi* gene, which is involved in all *K. pneumoniae* capsular types [121] codes for an outer membrane protein that attaches the capsule to the cell surface. This protein is useful in predicting the K type and classifying strains based on the capsule. The capsule has been associated with virulence and resistance [122]. Other genes, such as *wza* and *wzb*, are involved in exporting and assembling capsule polysaccharides. The KpnI/KpnJ system is activated when cell density reaches a certain threshold, resulting in the expression of genes involved in capsule synthesis and biofilm formation. *SdiA*, a QS regulator, suppresses fimbria expression, biofilm formation, and the production of QS-signaling molecules in *K. pneumoniae* [123]. These genes could serve as pharmacological targets to reduce or limit non-specific resistance mechanisms of *K. pneumoniae*, as previously suggested [124]. Additionally, Table 1 was added, which lists the main genes associated with the induction/inhibition processes of the QS state.

### 5.5. pH Conditions Regulate Antibiotic Resistance 

Another of the major homeostatic regulation mechanisms that has been related to antimicrobial resistance processes has been the range of pH values. Studies related to *K. pneumoniae* are scarce but should be included in this review. This is an analysis of bacteria with pathogenic potential in the urinary tract, including *K. pneumoniae*. The contribution between efflux systems was determined under acidic pH conditions, modifying their susceptibility to different antibiotics [141]. Using the same approach, the susceptibility of *K. pneumoniae* to ciprofloxacin and fosfomycin was also investigated [142]. In a separate study, researchers analyzed the impact of pH changes in the intestinal environment on *K. pneumoniae* resistance [143]. Additionally, a relevant study not specific to *K. pneumoniae* found that pH and oxygen exposure are key environmental factors affecting GRN and colistin resistance [4]. In this same sense, some studies have linked osmotic pressure and acidic conditions to the homeostatic regulation of resistance to aminoglycosides and tetracyclines [144]. 

## 6. Transcriptional Regulation and Cryptic Resistance Mechanisms 

This section explores the role of transcription factors (TFs) in resistance mechanisms that operate cryptically. While TFs participate in each of the homeostatic regulation processes in different ways and multiple genetic circuits, it is important to examine the specific role of different TFs in resistance mechanisms. Consequently, resistance mechanisms may be operating in the GRN in an unknown manner. Additionally, the proteins and circuits involved cannot be related to the resistance mechanisms already published and known in databases [145,146]. In *K. pneumoniae*, there are also some relevant studies to cite. LysR-type transcriptional regulators (LTTRs) are a family of DNA-binding proteins that regulate gene expression in prokaryotes. They play a key role in controlling several physiological processes, including metabolism, virulence, and stress response [147]. The diversity and complexity of LTTRs in *K. pneumoniae* highlight their importance in adapting and surviving in different environments, including their ability to colonize and infect humans. Proteins bind to specific regions of DNA known as regulator binding sites (RBSs) and can function as either activators or repressors of gene transcription [148]. A study conducted in *K. pneumoniae* investigated the Type VI Secretion System (T6SS) factor, which is associated with cell invasion, competition, and colonization processes. The study determined its relevance in resistance to carbapenem antibiotics [149]. Finally, some studies have used the complete analysis of the *K. pneumoniae* genome as an approach to elucidate cryptic mechanisms, although they are not focused on transcriptional regulation [150]. 

## 7. Moonlighting Proteins Associated with Resistance

Moonlighting proteins can perform more than one function, depending on the phy- siological state of the cell and their interaction profile. *Klebsiella* species have several moonlighting proteins, including enolase, which is involved in both glycolysis and bacterial adhesion [151]. Enolase interacts with plasminogen, a protein involved in blood clotting, which may help the bacteria evade the host’s immune system. Another example is the chaperone protein GroEL, which is involved in protein folding and plays a role in bacterial adhesion and invasion. It interacts with host cell surface receptors, allowing the bacteria to adhere to and invade host cells [152]. The significance of these proteins lies in their precise function in various homeostatic regulation mechanisms. Their functional profile is dependent on the physiological state of the GRN, which fluctuates differently under conditions of selection pressure with and without antibiotics. 

## 8. Conclusions 

In this review work, it can be concluded that the most general mechanisms of bacterial adaptation such as homeostatic regulation have a fundamental role as a central element and/or adjuvant in the processes of antimicrobial resistance. Assuming that the adaptive–evolutionary process has been constant in natural history, it can be inferred that the emergence of resistance will continue under any clinical and/or ecological circumstance. Therefore, it is necessary to consider some initial approaches for controlling the phenomenon of antimicrobial resistance and its impact on the gene regulatory network (GRN) of *K. pneumoniae*. Based on the references cited in this study, several conclusions can be drawn regarding the relationship between resistance and homeostasis. (1) In general, it can be said that *K. pneumoniae* tends to adapt to any stressful ecosystem context. In a scenario involving antibiotics, the bactericidal effect on a biofilm can transition from susceptibility to a bacteriostatic state and progress to resistance. This change can be understood as a modification of two fundamental variables of the gene regulatory network (GRN) of *K. pneumoniae* and other clinically relevant bacteria. Firstly, (a) there is the genetic expression regulation, namely either the upregulation or downregulation of different relevant mechanisms related to resistance and homeostasis before and after exposure to the drug. Secondly, there is (b) the generation of genetic variability that can come from two large categories of mechanisms, namely (i) horizontal genetic transfer (HGT) phenomena whose intensity in terms of information flow increases under stress conditions [153,154] and (ii) the introduction of changes in the sequence (mutations) derived from high levels of oxidative stress [111]. All of these mechanisms are particularly important in *K. pneumoniae*, a microorganism in which hypermucoviscous phenotypes have been identified [155,156], and which has other peculiarities in the capsule and glycocalyx. The effects of the previously mentioned processes are especially intense in *K. pneumoniae*, with the fundamental objective being to achieve sub-lethal conditions in terms of antimicrobial concentration in a non-specific manner. (2). The process by which *K. pneumoniae* transitions from being susceptible to resistance can be summarized as follows. Under an intermediate state of bacteriostasis, homeostatic regulation is oriented in the gene regulatory network (GRN) to optimize a resistant phenotype. In this state, the antibiotic affects the general homeostatic process through the energy flow of the cell, which may affect general processes of bacterial molecular genetics such as replication, protein synthesis, and wall synthesis. Consequently, the demand for energy flow must be redirected to attenuate these cytotoxic effects. This overstimulates the electron transport [16] at the membrane level, resulting in the formation of reactive oxygen species. If attenuation is successful, the time during which selection pressure and bacteriostasis are maintained increases. This triggers a response in the GRN through more specific and energy-efficient mechanisms. As a result, the bacterial population can allocate some of its energy towards reproduction, thereby articulating the resistance phenotype. Furthermore, it is crucial to acknowledge the versatility and genomic diversity of *K. pneumoniae*. This information is of particular importance for upcoming research aimed at understanding and combatting this microorganism effectively. A more profound comprehension of these mechanisms may offer insights into the development of efficient strategies for preventing and treating *K. pneumoniae* infections, as well as addressing the increasing threat of antimicrobial resistance worldwide. 

## 9. Recommendations for Future Research 

This study’s approach enables the configuration of a horizon of new research questions to generate a more comprehensive understanding of the resistance phenomenon and cellular homeostasis. This understanding must inevitably involve the analysis of the genomic network (GRN) and its adaptive response under stress conditions. The use of various omics techniques, such as interactomics, proteomics, and metabolomics, to analyze the contributions of mechanisms and processes that can provide a deeper understanding of interaction phenomena in gene regulatory networks (GRNs). This understanding can lead to the optimization of adaptive responses, which all organisms in the biosphere pursue to increase their biological effectiveness, as measured by growth or biomass generation and successful reproduction.

## Figures and Tables

**Figure 1 antibiotics-13-00490-f001:**
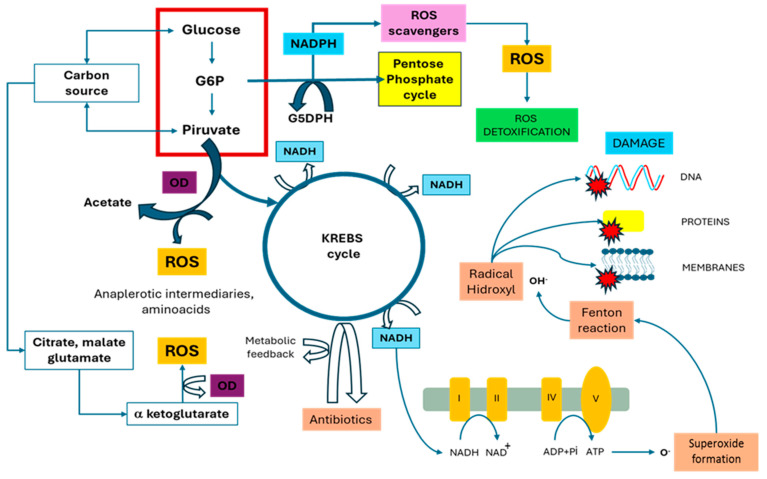
The image shows that the production of reactive oxygen species (ROS) occurs through the primary metabolism and the adjuvant phenomenon in the presence of antibiotics. This can lead to an increase in the concentration of ROS and cellular damage to nucleic acids, membranes, and proteins. The Roman numerals represent the domains of the electron transport chain where reactive oxy-gen species are produced (OD) Oxidative decarboxylation; (NADPH) nicotinamide adenine dinucleotide phosphate; (G6P) glucose 6 phosphate; (G5DP) glucose 5 diphosphate.

**Figure 2 antibiotics-13-00490-f002:**
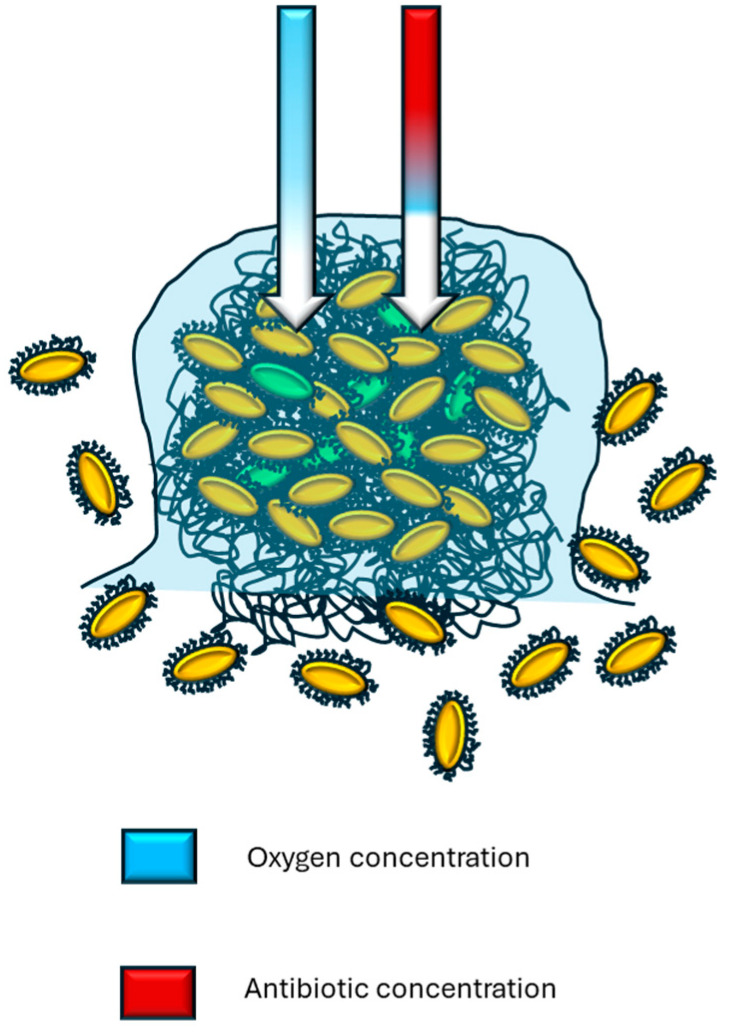
The image highlights that the bacterial organization in the form of a biofilm facilitates the attenuation of environmental variables due to the retention that occurs in the layers of lipopolysaccharides present throughout the structure. The image shows how antibiotics and oxygen are distributed heterogeneously by establishing differential concentration gradients at the periphery of the biofilm with respect to the interior, which generates phenotypic diversity among the bacterial cells of the biofilm.

**Figure 3 antibiotics-13-00490-f003:**
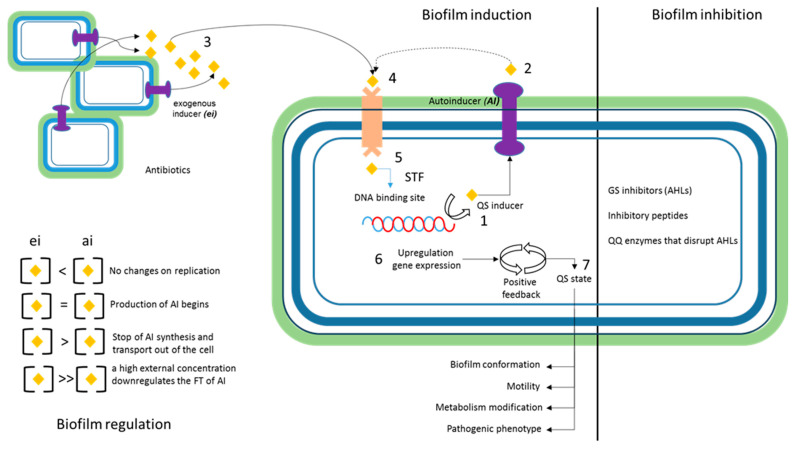
General mechanism of the QS physiological state biofilm. Induction: 1. Basal production of auto-inducers (AIs). 2. Presence of AIs on the cell exterior 3. Production of exogenous inducers 4. Perception of Ais and EIs (exogenous inducers) 5. AI/EI concentration balance ((SFT) specific transcription factors, transfer information) 6. Overexpression of QS inducers and change in the regulation of expression. Biofilm inhibition: (QQE); quorum quenching enzymes. N-acyl-homoserine lactones (AHLs). 7. The cellular physiology processes that are activated in the QS state are listed.

**Table 1 antibiotics-13-00490-t001:** Lists the main genes and proteins associated with the induction and inhibition of the quorum sensing mechanism. Some of the abbreviations included are the following: homoserine lactone acylase (AHL-ase), quorum quenching (Qq), acyl-homoserine lactones (AHLs).

Inhibitor	Function	Reference
**Furanones**	They inhibit the production of auto-inducers by interrupting signaling, which prevents bacteria from producing and detecting the auto-inducing molecules necessary to coordinate their behavior, reducing cell density-dependent gene expression and that related to virulence.	[125,126]
**(AHL-ase)**	Enzyme that is responsible for degrading autoinduction molecules (AHLs) produced by Gram-negative bacteria and, therefore, their communication.	[125,126,127]
**Qq**	Mechanism that involves the enzymatic degradation of signaling molecules used by bacteria to synchronize their behavior within communities. It prevents the accumulation of signaling molecules in the medium, which inhibits QS. It is classified as follows: class I includes lactonases, acylases and paraoxonases. Class II refers to oxidoreductases.	[127,128,129,130,131,132]
**Inductor**		
**(AHLs)**	Acyl-homoserine lactones (AHLs) are signaling molecules in the quorum sensing system, which play a role in the coordination of various bacterial processes; Inside the cell, AHLs bind to LuxR-type receptor proteins, forming a transcription activating complex.	[133,134,135]
**AlQS System**	The Auto-Inducer Quorum Sensing System is an intercellular communication mechanism that allows bacteria to coordinate their behavior based on population density. This system involves the synthesis and detection of signaling molecules (auto-inducers), which bind to specific receptors on the bacterial surface when they reach a critical concentration, activating the expression of specific genes that regulate processes such as biofilm production, virulence, etc	[133,134,135]
**Lux regulators**	The Lux-type regulator family is a group of proteins essential for cellular communication and the coordination of events depending on the density of the bacterial population. They are classified based on their structure and function, with LuxR being the main regulator. - LuxR: main transcriptional regulator that binds to auto-inducing molecules and activates the expression of specific genes. - LuxI: enzyme that synthesizes auto-inducing molecules, such as acyl-homoserine lactones (AHLs). - LuxS: enzyme involved in the synthesis of signaling molecules, such as autoinducer-2 (AI-2), which participates in intercellular communication	[125,135,136,137,138,139]
**SdiA**	Transcriptional regulator that belongs to the family of LuxR-type regulators, acting by suppressing the expression of fimbriae and biofilm formation, in addition to regulating the expression of genes related to virulence.	[136,139]
**di-GMPc**	Second intracellular messenger that plays a crucial role in regulating biofilm formation. This molecule is involved in signal transduction and controls the expression of genes related to the production of exopolysaccharides and biofilm formation in bacteria.	[125,139,140]
**Nitric oxide**	Signaling molecule that participates in intercellular communication and in the regulation of various physiological processes in bacteria. In the context of quorum sensing, nitric oxide can act as an environmental signal that modulates biofilm formation and bacterial motility in response to specific conditions.	[125,139,140]

## Data Availability

Not applicable.

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
