# Peer review of "Relevance of the Adjuvant Effect between Cellular Homeostasis and Resistance to Antibiotics in Gram-Negative Bacteria with Pathogenic Capacity: A Study of Klebsiella pneumoniae"

_antibiotics, 2024, doi:10.3390/antibiotics13060490_

Round 1

Reviewer 1 Report

Comments and Suggestions for Authors

File attached

Reviewer 2 Report

Comments and Suggestions for Authors

The authors provide a review examining the relevance of homeostatic regulatory mechanisms in antimicrobial resistance mechanisms. They focused on interactions in the cellular physiology of pathogenic bacteria, particularly gram-negative bacteria, specifically Klebsiella pneumoniae. The research addresses the relevance of homeostatic regulatory mechanisms in antimicrobial resistance, particularly focusing on Klebsiella pneumoniae and its association with healthcare-associated infections. Specifically, the study aims to understand how homeostatic regulation processes contribute to the emergence and maintenance of antimicrobial resistance in K. pneumoniae.

The research delves into the intricate mechanisms of cellular homeostasis in bacteria, emphasizing their role in resisting antibiotics. It explores how alterations in metabolic pathways and the formation of biofilms contribute to antibiotic resistance, particularly in K. pneumoniae. The study also investigates the genetic circuits involved in homeostatic regulation and resistance emergence, offering insights into potential targets for future research. The exploration of various homeostatic regulation mechanisms, such as cellular efflux, heat shock proteins, response to oxygen availability and reactive oxygen species (ROS), quorum sensing, pH conditions, transcriptional regulation, and moonlighting proteins, in the context of antimicrobial resistance.

Compared to other published material, this research provides a comprehensive examination of the interplay between homeostatic regulation mechanisms and antibiotic resistance in K. pneumoniae. It offers novel perspectives on how cellular physiology and genetic circuits influence resistance phenotypes, paving the way for the development of innovative strategies to combat antimicrobial resistance. Additionally, the paper highlights the interconnectedness of these mechanisms and their adaptive significance in the context of antibiotic exposure, contributing to a deeper understanding of antimicrobial resistance in K. pneumoniae.

In the case of a review manuscript, there is nothing that can be added about a control group or any improvement in methodology.

The conclusions drawn from the evidence presented are generally consistent, highlighting the significance of homeostatic regulation mechanisms in antibiotic resistance.The study successfully addresses the main question posed regarding the relationship between antimicrobial resistance mechanisms and cellular homeostasis in K. pneumoniae. By synthesizing findings from various studies and integrating them into a cohesive framework, the conclusions provide valuable insights into the adaptive responses of K. pneumoniae to antibiotic exposure and the implications for antimicrobial resistance. However, specific experiments directly addressing these conclusions were not outlined in the text, suggesting that the conclusions are drawn based on the collective evidence from the literature reviewed.

The references are appropriate.

Nothing to declare regarding the figures.

Reviewer 3 Report

Comments and Suggestions for Authors

I have read with interest the manuscript submitted by Lopez-Perez et al, since AMR  represents a global concern.

I have some comments to be addressed in order to improve the quality of the manuscript:

- the abstract should be completely rephrased.

- many phrases are ambiguous and hard to understand. Maybe a clinician should revise the manuscript, in order to fix these discrepancies. 

- many phrases inserted are not supported by a reference.

- all abbreviations used in the text or figures should be described at first use/legends.

- I recommend a more methodized approach (with clear subsections and organized information) for the entire manuscript; just an example: when discussing biofilm, the definition should be placed in the beginning, not later.

- Figure 2 is not so clear -the message meant to be provided is not understandable.

- please replace Enterobacteriaceae with Enterobacterales

- when discussing about K. pneumoniae's resistance mechanisms, it is mandatory to include the enzyme production, such as ESBL, carbapenemases, or harboring genes such as mcr, etc (information important for clinicians).

- "A study published in K. pneumoniae found that...." ?

- the conclusion section is too long.

- the reference list is not edited according to the mdpi pattern.

Comments on the Quality of English Language

A great amount of information should be rephrased.

There are also punctuation issues. 

Round 2

Reviewer 3 Report

Comments and Suggestions for Authors

I appreciate the author's efforts in addressing my comments. The quality of the manuscript has improved. My only minor comments are:

Make sure that all Latin bacterial names are italicized

row 37 - avoid the double use of the word history

"A study published in K. pneumoniae found that" - my problem with this phrase was that the study was not published in K. pneumoniae, but about this microorganism.

Table 1 - the title should be shorter and the legend should be placed under the table

row 629 - the word news is not a more appropriate term

The phrases are significantly shorter. Further re-arrangements such as the division of the bulky one-page-long paragraphs into more organized, fragmented ones are advised.

The review is of great importance, therefore the presentation of the research in a clear and organized manner would improve the reader's interest (and also the citations). I only pointed out the clinician's point of view, to make the article appealing to this category of readers.

Best regards,

Comments on the Quality of English Language

minor editing

Author Response

Please see in the attachment.
